# Beamsteering-Aware Power Allocation for Cache-Assisted NOMA mmWave Vehicular Networks

Wei Cao [1,†] , Jinyuan Gu [2,†] , Xiaohui Gu [1,*] and Guoan Zhang [1]

1    School of Information Science and Technology, Nantong University, Nantong 226019, China;
     2110310037@stmail.ntu.edu.cn (W.C.); gzhang@ntu.edu.cn (G.Z.)
2    Kangda College, Nanjing Medical University, Lianyungang 222000, China; gujinyuan@njmu.edu.cn
*    Correspondence: gu.xh@ntu.edu.cn
†    These authors contributed equally to this work.

**Abstract:** Cache-enabled networks with multiple access (NOMA) integration have been shown to decrease wireless network traffic congestion and content delivery latency. This work investigates optimal power control in cache-assisted NOMA millimeter-wave (mmWave) vehicular networks, where mmWave channels experience double-Nakagami fading and the mmWave beamforming is subjected to beamsteering errors. We aim to optimize vehicular quality of service while maintaining fairness among vehicles, through the maximization of successful signal decoding probability for paired vehicles. A comprehensive analysis is carried out to understand the decoding success probabilities under various caching scenarios, leading to the development of optimal power allocation strategies for diverse caching conditions. Moreover, an optimal power allocation is proposed for the single-antenna case, for exploiting the cached data as side information to cancel interference. The robustness of our proposed scheme against variations in beamforming orientation is assessed by studying the influence of beamsteering errors. Numerical results demonstrate the effectiveness of the proposed cache-assisted NOMA scheme in enhancing cache utility and NOMA efficiency, while underscoring the performance gains achievable with larger cache sizes.

**Keywords:** vehicular networks; nonorthogonal multiple access; mmWave communications; spectral efficiency; power allocation

## 1. Introduction

In recent decades, wireless communication technologies have evolved rapidly in response to an escalating demand for enhanced data rates, minimized latency, and reliable connectivity. The ambition of fifth-generation (5G) mobile networks and beyond is to address the surging increase of connected devices, data traffic, and diverse quality-of-service (QoS) requirements. Nonorthogonal multiple access (NOMA) has surfaced as an auspicious multiple-access technique to amplify spectral efficiency and network capacity, presenting significant advancements compared to its orthogonal multiple access (OMA) counterpart [1].

The rapid proliferation of multimedia content and mobile edge computing necessitates effective content delivery and caching strategies. Integrating caching and NOMA techniques can enhance system performance and reduce latency in vehicular networks, thereby relieving pressure on backhaul links and improving overall network efficiency [2,3]. Although cache-aided NOMA holds the promise of improving spectral efficiency and QoS from a theoretical standpoint, its practical benefits in terms of managing sporadic network connectivity and reducing latency by serving frequently requested data locally are still underexplored. These benefits are particularly significant for vehicular networks, characterized by high mobility and resource constraints, which can leverage caching to manage fluctuating network conditions and lessen backhaul link load, while NOMA can be used to increase user capacity.

As another pivotal technology in the communication domain, millimeter-wave (mmWave) communication is expected to meet the high data rate requirements of 5G networks and beyond [4–6]. Despite its potential to accommodate multigigabit data rates through the vast available bandwidth in the mmWave frequency band [7], challenges persist due to severe path loss and sensitivity to blockage characteristics of mmWave signals. These issues underscore the necessity for sophisticated beamforming techniques and multiantenna systems to ensure reliable communication [8].

Motivated by these challenges and opportunities, we propose a cache-assisted NOMA framework for mmWave vehicular networks in this paper. Our framework aims to achieve a balance between optimizing decoding performance and enhancing system efficiency. The main contributions of this paper are as follows:

- The development of a novel cache-assisted NOMA framework for mmWave vehicular networks that exploits the synergies between mmWave beamforming and cache-assisted NOMA. This framework promises substantial improvements in communication efficiency and system reliability. We incorporated the probabilistic line-of-sight (LoS) path model and the double-Nakagami fading model into our framework to accurately simulate real-world propagation conditions.
- A thorough analysis of decoding success probabilities under diverse caching conditions. Our approach addresses multiple caching scenarios and advocates for fairness among paired vehicles by formulating an optimization problem aimed at maximizing the product of their individual decoding success probabilities. We also devised optimal power allocation strategies for each unique caching condition.
- A rigorous numerical analysis demonstrating the robustness of our proposed cache-assisted NOMA framework against beamsteering errors. Our findings underscore the superiority of our scheme over the traditional NOMA counterpart and show how augmenting cache size can lead to performance improvement.

The rest of the paper is organized as follows: Section 2 provides a review of related work in the field of cache-assisted NOMA and mmWave vehicular networks. The system model is presented in Section 3 followed by the problem formulation, while Sections 4 and 5 describe the proposed cache-assisted NOMA scheme for single- and multiple-antenna cases, respectively. Section 6 presents the simulation results and performance analysis, and finally, Section 7 concludes the paper.

## 2. Related Work

In recent years, the integration of caching and NOMA techniques, along with the implementation of mmWave communication in vehicular networks, has attracted considerable attention from researchers. In this section, we provide an overview of the related work in the areas of cache-assisted NOMA and mmWave vehicular networks.

### 2.1. Cache-Aided NOMA

Caching is widely acknowledged as an effective method for alleviating backhaul link loads and reducing latency in wireless networks. Combining caching with NOMA has shown potential for further enhancing system performance. For instance, Lin et al. [9] investigated the impact of caching on the performance of a NOMA-based cellular network, focusing on the effects of various cache configurations and decoding orders on the achievable rate region. They also studied optimal power and rate allocation for minimizing delivery time. The authors of [10] examined the use of NOMA and coded multicasting in cache-assisted networks and proposed a hybrid delivery scheme that selects either NOMA or coded multicasting depending on the channel conditions of paired users in each resource block.

In [11], the authors presented cache-assisted NOMA as a facilitator for vehicular networks, addressing challenges stemming from the growing need for high-quality multimedia services and the rise of the Internet of Things (IoT) in vehicular network contexts. Their findings revealed that cache-assisted NOMA outperforms traditional NOMA techniques

and offers performance benefits in vehicular networks. Zhang et al. [12] explored the ideal power control and successive interference cancellation (SIC) order selection issue in cache-assisted NOMA vehicular networks. Their objective was to determine the optimal SIC ordering to optimize the lowest achieved average outage data rate.

### 2.2. mmWave Vehicular Networks

The application of mmWave communication in vehicular networks has emerged as a promising approach to achieve high data rates and low latency. Numerous studies have explored the performance of mmWave-based vehicular communication systems while addressing challenges related to the propagation characteristics of mmWave signals. For example, Li et al. [13] provided a comprehensive survey of opportunities and technologies supporting mmWave communications in mobile scenarios, discussing challenges, opportunities, and potential solutions concerning channel modeling, channel estimation, antiblockage, and capacity enhancement.

Moreover, Zhang et al. [14] evaluated the performance of beamforming techniques in mmWave vehicular networks and proposed an adaptive beamforming scheme for message dissemination in mmWave vehicle-to-vehicle (V2V) communications. Mahabal et al. [15] assessed the impact of vehicle dynamics and beamforming on direct V2V communication in a highway environment, analyzing the joint effect of antenna elements and the number of bits in the digital phase shifter on link coverage probability. In [16], the authors investigated the beam management problem in urban vehicular networks and devised a graph-based model to capture system characteristics resulting from rapid user mobility. To cater to future vehicular wireless applications requiring high-throughput ultrareliable low-latency communications, Ref. [17] introduced a hybrid network design combining mmWave and sub-6 GHz communications for vehicle-to-everything (V2X) networks.

### 2.3. Cache-Aided NOMA in mmWave Vehicular Networks

Despite the growing interest in cache-assisted NOMA and mmWave vehicular networks, only a limited number of studies have explored the integration of these technologies. In [18], the authors proposed a cache-enabled NOMA scheme for vehicular communication, employing power-domain NOMA and content caching to enhance system capacity. However, this work did not consider the impact of beamforming and beamsteering errors on system performance.

To summarize, while the integration of caching and NOMA techniques has been investigated in various contexts, the incorporation of cache-assisted NOMA and mmWave communication in vehicular networks, particularly considering the impact of beamsteering errors and diverse caching conditions, has not been extensively explored. This paper seeks to fill this gap by proposing and analyzing a cache-assisted NOMA scheme for multiantenna mmWave vehicular networks under diverse system conditions.

## 3. System Model

We explore vehicular networks where a base station (BS) simultaneously facilitates communication with two single-antenna vehicles, labeled as $V_1$ and $V_2$, as illustrated in Figure 1. To accommodate this dual communication, the BS operates a two-beam mmWave transmitter [19], which comprises separate antenna arrays. Specifically, the first antenna array is devoted to serving $V_1$, while the second array is assigned to $V_2$. Remarkably, a pioneering prototype of an mmWave antenna system, referenced in [20], features two distinct mmWave antenna arrays in a $1 \times 16$ configuration. These arrays are integrated into a Samsung mobile device, positioned at its upper and lower extremities.

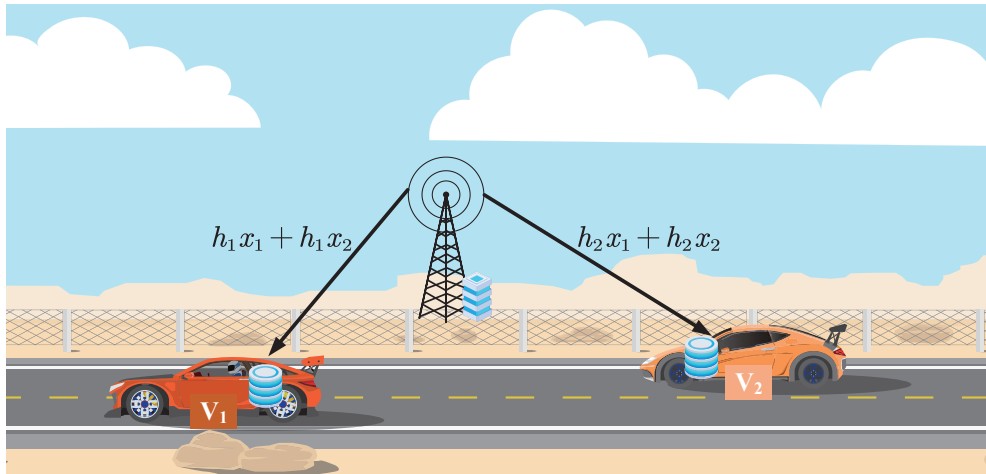

**Figure 1.** A cache-assisted NOMA vehicular framework, with a BS simultaneously catering to vehicles $V_1$ and $V_2$.

### 3.1. mmWave Channel Model

The distances and channels between the BS and vehicle $k \in \{1, 2\}$ are denoted by $d_k$ and $h_k$, respectively. To model mmWave links that exhibit high sensitivity to blocking effects, we consider two different sets of parameters. We define a deterministic function, $P_{\text{LoS}}(d_k) = (-d_k/C) \in [0, 1]$, that decreases with distance, to describe the probability of a link of length $d_k$ being LoS. According to the methodology provided in [19], the channel coefficient for an arbitrary mmWave link between the BS and vehicle $k$ is denoted as $h_k = \tilde{h}_k \sqrt{G_k L_k}$, with $\tilde{h}_k$, $G_k$, and $L_k$ representing the channel fading coefficient, diversity gain, and path loss, respectively. Moreover, the symbols used in this paper are listed in Table 1.

**Table 1.** List of symbols.

| Symbol | Description |
|---|---|
| $V_k$ | Vehicle $k$, where $k \in \{1, 2\}$ |
| $d_k$ | Distance between BS and vehicle $k$ |
| $h_k$ | Channel between BS and vehicle $k$ |
| $\alpha_{\text{L}}, \alpha_{\text{N}}$ | Path loss exponents for LoS and NLoS |
| $L_k$ | Path loss of link of length $d_k$ |
| $C_{\text{L}}, C_{\text{N}}$ | Intercepts for LoS and NLoS path losses |
| $G_k$ | Total diversity gain for vehicle $k$ |
| $M_k, m_k$ | Primary and secondary lobe gains for vehicle $k$ |
| $\theta_k$ | Beamwidth for vehicle $k$ |
| $\xi_k$ | Beamsteering error for vehicle $k$ |
| $\tilde{h}_k$ | Fading amplitude of the mmWave link for vehicle $k$ |
| $\mathcal{F}$ | Database of popular files |
| $T$ | Total number of files, corresponding to the BS's caching capacity |
| $q_t$ | Popularity of file $t$ |
| $f_k$ | File cached by vehicle $k$ |
| $x_k$ | Signal associated with the request of $V_k$ |
| $P$ | BS's transmit power |
| $\alpha$ | Power proportion allocated to $V_1$ |
| $G_{k,f}, G_{k,s}$ | Cumulative diversity gains for vehicle $k$ from primary and secondary arrays |
| $y_k$ | Signal received by $V_k$ |
| $\sigma_k$ | Variance of zero-mean Gaussian noise component at the receiver of $V_k$ |

Path losses for the LoS and non-line-of-sight (NLoS) links are denoted by $L_k^{(L)}$ and $L_k^{(N)}$, respectively. Different path loss exponents, $\alpha_L$ for LoS and $\alpha_N$ for NLoS, are employed. For a link of length $d_k$ from the BS to vehicle $k$, its path loss $L_k$ is calculated by

$$L_k(d_k) = \begin{cases} C_L d_k^{-\alpha_L} & w \cdot p \cdot P_{\text{LoS}}(d_k) \\ C_N d_k^{-\alpha_N} & w \cdot p \cdot 1 - P_{\text{LoS}}(d_k) \end{cases} \tag{1}$$

where $C_L$ and $C_N$ signify the intercepts for LoS and NLoS path losses, respectively.

In this study, our primary aim is to investigate the impact of caching on the performance of cache-aided NOMA mmWave vehicular networks, particularly focusing on the effects of beamsteering errors and different caching conditions. For this purpose, we chose to utilize a simpler power allocation scheme. The rationale behind this choice is twofold: Firstly, it allows us to keep our focus on the main variables we wish to study, namely the beamsteering errors and different caching conditions. Secondly, it allows for a clear understanding of these specific effects without the additional complexities that joint beamforming and power allocation would introduce.

In terms of antenna modeling, we assume that the total diversity gain $G_k$ is determined using the sectored-pattern antenna model, as widely accepted in the literature [21,22]. This model approximates the real array beam pattern with a step function, featuring a constant primary lobe gain $M_k$ over the beamwidth $\theta_k$ and a constant secondary lobe gain $m_k$. Assuming a beamsteering error, denoted by a symmetric random variable $\xi_k$, for the primary lobe pointed towards $V_k$ and following [21], the probability mass function of the diversity gain for $V_k$ is modeled as

$$G_k = \begin{cases} M_k & w \cdot p \cdot F_{|\xi_k|}(\theta_k/2) \\ m_k & w \cdot p \cdot 1 - F_{|\xi_k|}(\theta_k/2) \end{cases} \tag{2}$$

where $F_{|\xi_k|}(\theta_k/2)$ denotes the probability that $G_k = M_k$ when $|\xi_k| < \theta_k/2$.

We assume a quasi-static fading channel for the connection between the BS and each vehicle, typically observed during peak traffic hours [23]. To model the fading characteristics in vehicular communication systems accurately, we represent the fading amplitudes of the mmWave links using the product of two independent Nakagami-$m$ distributions: $\left|\tilde{h}_k^{(N)}\right| \sim \mathcal{N}^2\left(\left(m_{1,k}^{(N)}, \Omega_{1,k}^{(N)}\right), \left(m_{2,k}^{(N)}, \Omega_{2,k}^{(N)}\right)\right)$ and $\left|\tilde{h}_k^{(L)}\right| \sim \mathcal{N}^2\left(\left(m_{1,k}^{(L)}, \Omega_{1,k}^{(L)}\right), \left(m_{2,k}^{(L)}, \Omega_{2,k}^{(L)}\right)\right)$ [24]. Here, $m_{1,k}^{(N)}, m_{2,k}^{(N)}, m_{1,k}^{(L)}, m_{2,k}^{(L)}$ represent the shape parameters for the different Nakagami-$m$ distributions that represent the fading amplitude of the mmWave links for vehicle $k$ under NLoS and LoS conditions, respectively. $\Omega_{1,k}^{(N)}, \Omega_{2,k}^{(N)}, \Omega_{1,k}^{(L)}, \Omega_{2,k}^{(L)}$ represent the spread parameters for the different Nakagami-$m$ distributions that model the fading amplitude of the mmWave links for vehicle $k$ under NLoS and LoS conditions, respectively. As a result, the probability density function (PDF) of $\left|\tilde{h}_k\right|$ is expressed as follows:

$$f_{|\tilde{h}_k|}(y) = \frac{2G_{0,2}^{2,0}\left(\frac{m_{1,k}^{(B)} m_{2,k}^{(B)} y^2}{\Omega_{1,k}^{(B)} \Omega_{2,k}^{(B)}} \middle| \begin{array}{c} \overline{\phantom{xx}} \\ m_{1,k}^{(B)}, m_{2,k}^{(B)} \end{array}\right)}{y\Gamma\left(m_{1,k}^{(B)}\right)\Gamma\left(m_{2,k}^{(B)}\right)}, \tag{3}$$

where $B \in \{L, N\}$, $\Gamma(\cdot)$ and $G_{m,n}^{p,q}(\cdot)$ denote Euler's gamma function and Meijer-G function, respectively [25]. The double Nakagami-$m$ distribution extends the double Rayleigh distribution, which is included as a special case [26], by acting as a cascaded fading model. Additionally, a double Gamma distribution is followed by the square of a double Nakagami-$m$ distributed random variable, $\left|\tilde{h}_k\right|^2$. The density of this distribution can be expressed through a standard transformation of random variables as follows:

$$f_{|\tilde{h}_k|^2}(y) = \frac{2y^{\frac{m_{1,k}^{(\mathcal{B})} - m_{2,k}^{(\mathcal{B})}}{2} - 1}}{\Gamma\left(m_{1,k}^{(\mathcal{B})}\right)\Gamma\left(m_{2,k}^{(\mathcal{B})}\right)\left(\frac{\Omega_{1,k}^{(\mathcal{B})}\Omega_{2,k}^{(\mathcal{B})}}{m_{1,k}^{(\mathcal{B})} m_{2,k}^{(\mathcal{B})}}\right)^{\frac{m_{1,k}^{(\mathcal{B})} + m_{2,k}^{(\mathcal{B})}}{2}}} \cdot \mathcal{K}_{m_{1,k}^{(\mathcal{B})} - m_{2,k}^{(\mathcal{B})}}\left(2\sqrt{\frac{m_{1,k}^{(\mathcal{B})} m_{2,k}^{(\mathcal{B})}}{\Omega_{1,k}^{(\mathcal{B})}\Omega_{2,k}^{(\mathcal{B})}}y}\right). \quad (4)$$

In the context of the modified Bessel function of the second kind, $\mathcal{K}_k(\cdot)$ represents the function with an order $n \in \mathbb{R}$ [25]. Hence, the associated cumulative distribution function (CDF) can be expressed as

$$F_{|\tilde{h}_k|^2}(y) = \frac{G_{1,3}^{2,1}\left(\frac{m_{1,k}^{(\mathcal{B})} m_{2,k}^{(\mathcal{B})} y}{\Omega_{1,k}^{(\mathcal{B})}\Omega_{2,k}^{(\mathcal{B})}} \middle| \begin{matrix} 1 \\ m_{1,k}^{(\mathcal{B})}, m_{2,k}^{(\mathcal{B})}, 0 \end{matrix}\right)}{\Gamma\left(m_{1,k}^{(\mathcal{B})}\right)\Gamma\left(m_{2,k}^{(\mathcal{B})}\right)}. \quad (5)$$

### 3.2. Caching Model

Consider a finite database of popular files, denoted by $\mathcal{F} \triangleq \{F_1, F_2, \ldots, F_T\}$. These files are accessible to vehicles, and the total number of these files, represented by $T$, corresponds to the BS's caching capacity. File popularity is modeled using the Zipf distribution [27], primarily due to its efficacy in representing the uneven distribution of file popularity in real-world scenarios. The Zipf distribution is often used in communication networks to model file popularity due to its 'heavy-tailed' property, wherein a small number of files are extremely popular, while the majority of files are accessed infrequently. This skewness in file popularity distribution is effectively captured by the skewness control parameter, $\zeta > 0$.

The probability associated with the popularity of file $F_t \in \mathcal{F}$ is given by

$$q_t = \frac{1}{t^{\zeta} \sum_{i=1}^{T} \frac{1}{i^{\zeta}}}, \ t = 1, \ldots, T, \quad (6)$$

subject to the condition $q_1 > q_2 > \cdots > q_T > 0$ and $\sum_{t=1}^{T} q_t = 1$. The skewness parameter, $\zeta$, controls the rate at which popularity decreases. A larger $\zeta$ results in a few extremely popular files and many rarely accessed files, while a smaller $\zeta$ signifies a more uniform distribution of file popularity. This parameter provides the flexibility to represent various realistic scenarios of file popularity, making it a versatile choice for our model. During off-peak hours, vehicles $V_1$ and $V_2$ cache the files $f_1 \in \mathcal{F}$ and $f_2 \in \mathcal{F}$, respectively. Given the popularity profile, the optimal caching strategy is to store files in descending order of their popularity.

During the request phase, vehicles $V_1$ and $V_2$ are assumed to request files $f_1$ and $f_2$, respectively, without loss of generality. We assume that the BS has complete knowledge about $f_1$ and $f_2$, and both $V_1$ and $V_2$ are aware of the power allocation strategy implemented by the BS. The BS gathers the necessary data during the request phase, particularly when vehicles $V_1$ and $V_2$ disclose their cache statuses. The BS can communicate the power allocation scheme to $V_1$ and $V_2$ prior to file transmission.

### 3.3. Downlink NOMA Communication Model

Let the signals associated with the requests of $V_1$ and $V_2$ be denoted by $x_1$ and $x_2$, respectively. The BS's transmit power is represented by $P$, and the power proportion allocated to $V_1$ is signified by $\alpha \in (0, 1)$. Let $G_{k,f}$ and $G_{k,s}$ represent the cumulative diversity gains of the links from the BS's primary and secondary arrays to vehicle $k = 1, 2$, respectively. Then, the signals received by $V_1$ and $V_2$ are written as

$$y_1 = \sqrt{\alpha P G_{1,f} L_1} \tilde{h}_1 x_1 + \sqrt{(1-\alpha) P m_{1,s} L_1} \tilde{h}_1 x_2 + n_1, \tag{7}$$

$$y_2 = \sqrt{\alpha P m_{2,f} L_2} \tilde{h}_2 x_1 + \sqrt{(1-\alpha) P G_{2,s} L_2} \tilde{h}_2 x_2 + n_2, \tag{8}$$

where $\tilde{h}_{k,f}$ and $\tilde{h}_{k,s}$ represent the small-scale fading from the BS's first and second arrays to vehicle $k = 1, 2$, respectively. Additionally, $n_1$ and $n_2$ denote zero-mean Gaussian noise components at the receivers of $V_1$ and $V_2$, with variances $\sigma_1^2$ and $\sigma_2^2$, respectively.

It is assumed that $V_1$ and $V_2$ are located in the primary lobes of the first and second antenna arrays of the BS, respectively. Furthermore, $V_1$ and $V_2$ receive the secondary lobes of the second and first arrays, respectively. Therefore, the BS transmits the NOMA signal to $V_1$ using the primary lobe of the first array and the secondary lobe of the second array, and $V_2$ receives the NOMA signal from the BS through the primary lobe of the second array and the secondary lobe of the first array. The signal-to-interference-plus-noise ratios (SINRs) for $V_k$ directly decoding $x_k$ are

$$\text{SINR}_1^{(x_1)} = \frac{\alpha P G_{1,f} L_1 |\tilde{h}_1|^2}{(1-\alpha) P m_{1,s} L_1 |\tilde{h}_1|^2 + \sigma_1^2}, \tag{9}$$

$$\text{SINR}_2^{(x_2)} = \frac{(1-\alpha) P G_{2,s} L_2 |\tilde{h}_2|^2}{\alpha P m_{2,f} L_2 |\tilde{h}_2|^2 + \sigma_2^2}, \tag{10}$$

where $G_{1,f} = \left[ M_{1,f} F_{|\xi_f|} + m_{1,f} (1 - F_{|\xi_f|}) \right]$ represents the total diversity gain of the first array directed toward $V_1$. Here, $M_{1,s}$ denotes the diversity gain from the secondary lobe of the second array to $V_1$. Additionally, $G_{2,s} = \left[ M_{2,s} F_{|\xi_s|} + m_{2,s} (1 - F_{|\xi_s|}) \right]$ implies the total diversity gain of the second array directed toward $V_2$, while $m_{2,f}$ refers to the diversity gain from the secondary lobe of the first array to $V_2$.

When $V_k$ needs to perform SIC, the corresponding SINRs for $V_1$ decoding $x_2$ and $V_2$ decoding $x_1$ are

$$\text{SINR}_1^{(x_2)} = \frac{(1-\alpha) P m_{1,s} L_1 |\tilde{h}_1|^2}{\alpha P G_{1,f} L_1 |\tilde{h}_1|^2 + \sigma_1^2}, \tag{11}$$

$$\text{SINR}_2^{(x_1)} = \frac{\alpha P m_{2,f} L_2 |\tilde{h}_2|^2}{(1-\alpha) P G_{2,s} L_2 |\tilde{h}_2|^2 + \sigma_2^2}. \tag{12}$$

Following this, $V_k$ should eliminate cochannel interference and decode its desired signal. The corresponding SNRs are given by

$$\text{SNR}_1^{(x_1)} = \frac{\alpha P G_{1,f} L_1^{\mathcal{B}} |\tilde{h}_1^{\mathcal{B}}|^2}{\sigma_1^2}, \tag{13}$$

$$\text{SNR}_2^{(x_2)} = \frac{(1-\alpha) P G_{2,s} L_2^{\mathcal{B}} |\tilde{h}_2^{\mathcal{B}}|^2}{\sigma_2^2}. \tag{14}$$

## 4. Power Allocation for Single-Antenna Case

This section investigates power allocations for $x_1$ and $x_2$ when the BS is equipped with a single-antenna ($G_k = 1$). The insights obtained here can be extended to the multiantenna scenario.

### 4.1. Probability of Successful Decoding

The optimal power allocation can be straightforward in certain situations, while more complex cases require the resolution of optimization problems to determine the most

efficient strategy. We first describe the simpler scenarios and their associated optimal power allocation policies, then tackle more intricate cases:

(1) When local caches can fulfill the service requirements of both vehicles, over-the-air transmissions become unnecessary.

(2) If a single vehicle can meet its request using its cache, the total power $P$ is dedicated to vehicle $V_k$. The successful file decoding for $V_k$ depends on the following condition

$$\rho_k L_k^{\mathcal{B}} \left| \tilde{h}_k^{\mathcal{B}} \right|^2 \geq \epsilon_k, \tag{15}$$

where $\rho_k = \frac{P}{\sigma_k^2}$. The probability of success is then computed as

$$\Pr\left\{ \left| \tilde{h}_k^{\mathcal{B}} \right|^2 \geq \frac{\epsilon_k}{\rho_k L_k^{\mathcal{B}}} \right\} = \sum_{\mathcal{B} \in \{L,N\}} P_{d_k}(\mathcal{B})[1 - \Phi_k(\varphi_k)], \tag{16}$$

where

$$\Phi_k(\varphi_k) = \frac{G_{1,3}^{2,1}\left( \frac{m_{1,k}^{(\mathcal{B})} m_{2,k}^{(\mathcal{B})}}{\Omega_{1,k}^{(\mathcal{B})} \Omega_{2,k}^{(\mathcal{B})}} \varphi_k \middle| \begin{matrix} 1 \\ m_{1,k}^{(\mathcal{B})}, m_{2,k}^{(\mathcal{B})}, 0 \end{matrix} \right)}{\Gamma\left( m_{1,k}^{(\mathcal{B})} \right) \Gamma\left( m_{2,k}^{(\mathcal{B})} \right)}, \tag{17}$$

and $\varphi_k = \frac{\epsilon_k}{\rho_k L_k^{\mathcal{B}}}$. It should be noted that vehicle $k$ cannot discern if the BS-vehicle connection is LoS or NLoS; thus, both scenarios should be considered, given the LoS probability $P_{\text{LoS}}(d_k)$. In (16), $P_{d_k}(L) = P_{\text{LoS}}(d_k)$ and $P_{d_k}(N) = 1 - P_{\text{LoS}}(d_k)$.

(3) If both vehicles require the same file but neither has it cached, a single signal representing the file is transmitted with power $P$ to both vehicles. Both vehicles can successfully decode the file when

$$\rho_k L_k^{\mathcal{B}} \left| \tilde{h}_k^{\mathcal{B}} \right|^2 \geq \epsilon_{1,2}, \tag{18}$$

where $\epsilon_{1,2}$ denotes the threshold for the SINR of the file requested by both vehicles. Consequently, the probability of successful decoding can be represented as

$$\Pr\left\{ \left| \tilde{h}_1^{\mathcal{B}} \right|^2 \geq \frac{\epsilon_{1,2}}{\rho_1 L_1^{\mathcal{B}}}, |\tilde{h}_2^{\mathcal{B}}|^2 \geq \frac{\epsilon_{1,2}}{\rho_2 L_2^{\mathcal{B}}} \right\}$$
$$= \sum_{\mathcal{B} \in \{L,N\}} P_{d_1}(\mathcal{B}) \left[ 1 - \Phi_1\left( \frac{\epsilon_{1,2}}{\rho_1 L_1^{\mathcal{B}}} \right) \right] \sum_{\mathcal{B} \in \{L,N\}} P_{d_2}(\mathcal{B}) \left[ 1 - \Phi_1\left( \frac{\epsilon_{1,2}}{\rho_2 L_2^{\mathcal{B}}} \right) \right]. \tag{19}$$

The presented cases depict a variety of power allocation scenarios, shedding light on the optimal utilization of power resources, an essential aspect in enhancing vehicle service performance. We will now discuss four complex scenarios depicted in Figure 2, where power allocation necessitates optimization.

- *Case I:* $V_1$ holds a cached copy of $f_2$, while $V_2$ experiences a cache miss.
- *Case II:* $V_1$ faces a cache miss, while $V_2$ possesses a cached copy of $f_1$.
- *Case III:* Both $V_1$ and $V_2$ have cached copies of $f_2$ and $f_1$, respectively.
- *Case IV:* Both vehicles do not have any cached files for the upcoming signal.

We will determine the optimal power allocation for each scenario in the subsequent analysis.

In *Case I*, $V_1$ reduces interference in the NOMA signal by leveraging the cached $f_2$, enabling the vehicle to decode the intended file if the following condition holds:

$$\frac{\alpha P L_1^{\mathcal{B}} \left| \tilde{h}_1^{\mathcal{B}} \right|^2}{\sigma_1^2} \geq \epsilon_1. \tag{20}$$

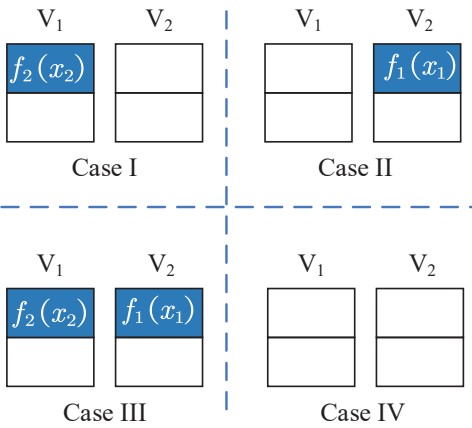

**Figure 2.** Four caching cases of paired vehicles.

The corresponding success probability is given by

$$P_{x_1}(\mathrm{I}) = \Pr\left\{ \left| \tilde{h}_1^{\mathcal{B}} \right|^2 \geq \frac{\epsilon_1}{\alpha \rho_1 L_1^{\mathcal{B}}} \right\} = \sum_{\mathcal{B} \in \{\mathrm{L,N}\}} P_{d_1}(\mathcal{B}) \left[ 1 - \Phi_1\left( \frac{\epsilon_1}{\alpha \rho_1 L_1^{\mathcal{B}}} \right) \right]. \tag{21}$$

For $V_2$, the parameter $\alpha$ yields two unique scenarios. If $0.5 \leq \alpha \leq 1$, $V_2$ needs to decode $f_1$, treating $f_2$ as interference. After decoding $f_1$, it is subtracted from the NOMA signal, and then $f_2$ is decoded. In contrast, when $0 < \alpha < 0.5$, $V_2$ decodes $f_2$ directly while treating $f_1$ as interference. These scenarios can be summarized as

- For the range $0.5 \leq \alpha \leq 1$, the following inequalities should hold:

$$\frac{\alpha P L_2^{\mathcal{B}} \left| \tilde{h}_2^{\mathcal{B}} \right|^2}{(1-\alpha) P L_2^{\mathcal{B}} \left| \tilde{h}_2^{\mathcal{B}} \right|^2 + \sigma_2^2} \geq \epsilon_1, \tag{22}$$

$$\frac{(1-\alpha) P L_2^{\mathcal{B}} \left| \tilde{h}_2^{\mathcal{B}} \right|^2}{\sigma_2^2} \geq \epsilon_2. \tag{23}$$

The success probability associated with these inequalities can be represented as

$$P_{x_2}(\mathrm{I}_1) = \Pr\left\{ \left| \tilde{h}_2^{\mathcal{B}} \right|^2 \geq \mathrm{I}_1^{\mathcal{B}} \right\} = \sum_{\mathcal{B} \in \{\mathrm{L,N}\}} P_{d_2}(\mathcal{B}) \left[ 1 - \Phi_2\left( \mathrm{I}_1^{\mathcal{B}} \right) \right], \tag{24}$$

where $\mathrm{I}_1^{\mathcal{B}} = \max\left\{ \frac{\epsilon_1}{[\alpha(1+\epsilon_1)-\epsilon_1]\rho_2 L_2^{\mathcal{B}}}, \frac{\epsilon_2}{(1-\alpha)\rho_2 L_2^{\mathcal{B}}} \right\}$.

- For the range $0 < \alpha < 0.5$, the following inequalities should hold:

$$\frac{(1-\alpha) P L_2^{\mathcal{B}} \left| \tilde{h}_2^{\mathcal{B}} \right|^2}{\alpha P L_2^{\mathcal{B}} \left| \tilde{h}_2^{\mathcal{B}} \right|^2 + \sigma_2^2} \geq \epsilon_2. \tag{25}$$

This results in the following success probability:

$$P_{x_2}(\mathrm{I}_2) = \Pr\left\{ \left| \tilde{h}_2^{\mathcal{B}} \right|^2 \geq \mathrm{I}_2^{\mathcal{B}} \right\} = \sum_{\mathcal{B} \in \{\mathrm{L,N}\}} P_{d_2}(\mathcal{B}) \left[ 1 - \Phi_2\left( \mathrm{I}_2^{\mathcal{B}} \right) \right], \tag{26}$$

where $\mathrm{I}_2^{\mathcal{B}} = \frac{\epsilon_2}{[1-(1+\epsilon_2)\alpha]\rho_2 L_2^{\mathcal{B}}}$.

In *Case II*, $V_2$ mitigates interference by utilizing cached content, allowing the desired signal to be decoded when

$$\frac{(1-\alpha)PL_2^{\mathcal{B}}|\tilde{h}_2^{\mathcal{B}}|^2}{\sigma_2^2} \geq \epsilon_2. \tag{27}$$

The respective success probability is calculated by

$$P_{x_2}(\text{II}) = \text{Pr}\left\{|\tilde{h}_2^{\mathcal{B}}|^2 \geq \frac{\epsilon_1}{(1-\alpha)\rho_2 L_2^{\mathcal{B}}}\right\} = \sum_{\mathcal{B}\in\{\text{L,N}\}} P_{d_2}(\mathcal{B})\left[1 - \Phi_2\left(\frac{\epsilon_2}{(1-\alpha)\rho_2 L_2^{\mathcal{B}}}\right)\right]. \tag{28}$$

Similarly to the prior case, the conditions for $V_1$ successfully decoding $f_1$ can be stated as

- For the range $0.5 \leq \alpha \leq 1$:

$$\frac{\alpha PL_1^{\mathcal{B}}|\tilde{h}_1^{\mathcal{B}}|^2}{(1-\alpha)PL_1^{\mathcal{B}}|\tilde{h}_1^{\mathcal{B}}|^2 + \sigma_1^2} \geq \epsilon_1. \tag{29}$$

The success probabilities are, respectively, given by

$$P_{x_1}(\text{II}_1) = \text{Pr}\left\{\left|\tilde{h}_1^{\mathcal{B}}\right|^2 \geq \text{II}_1^{\mathcal{B}}\right\} = \sum_{\mathcal{B}\in\{\text{L,N}\}} P_{d_1}(\mathcal{B})\left[1 - \Phi_1\left(\text{II}_1^{\mathcal{B}}\right)\right], \tag{30}$$

where $\text{II}_1^{\mathcal{B}} = \frac{\epsilon_1}{[(1+\epsilon_1)\alpha - \epsilon_1]\rho_1 L_1^{\mathcal{B}}}$.

- For the range $0 < \alpha < 0.5$, the following inequalities should hold:

$$\frac{(1-\alpha)PL_1^{\mathcal{B}}|\tilde{h}_1^{\mathcal{B}}|^2}{\alpha PL_1^{\mathcal{B}}|\tilde{h}_1^{\mathcal{B}}|^2 + \sigma_1^2} \geq \epsilon_2, \tag{31}$$

$$\frac{\alpha PL_1^{\mathcal{B}}|\tilde{h}_1^{\mathcal{B}}|^2}{\sigma_1^2} \geq \epsilon_1. \tag{32}$$

The success probability associated with these inequalities can be represented as

$$P_{x_1}(\text{II}_2) = \text{Pr}\left\{\left|\tilde{h}_1^{\mathcal{B}}\right|^2 \geq \text{II}_2^{\mathcal{B}}\right\} = \sum_{\mathcal{B}\in\{\text{L,N}\}} P_{d_1}(\mathcal{B})\left[1 - \Phi_1\left(\text{II}_2^{\mathcal{B}}\right)\right], \tag{33}$$

where $\text{II}_2^{\mathcal{B}} = \max\left\{\frac{\epsilon_2}{[1-\alpha(1+\epsilon_2)]\rho_1 L_1^{\mathcal{B}}}, \frac{\epsilon_1}{\alpha\rho_1 L_1^{\mathcal{B}}}\right\}$.

In *Case III*, both vehicles leverage their cached content to mitigate the interference stemming from the NOMA signal. This means that the conditions needed to successfully decode the intended signal do not depend on the value of $\alpha$ relative to 0.5. We can formally express the requirements for successfully decoding $f_1$ and $f_2$ at $V_1$ and $V_2$ as follows:

$$\frac{\alpha PL_1^{\mathcal{B}}|\tilde{h}_1^{\mathcal{B}}|^2}{\sigma_1^2} \geq \epsilon_1, \tag{34}$$

$$\frac{(1-\alpha)PL_2^{\mathcal{B}}|\tilde{h}_2^{\mathcal{B}}|^2}{\sigma_2^2} \geq \epsilon_2. \tag{35}$$

The probability of success for this case can be denoted as

$$P_{x_1}(\text{III}) = P_{x_1}(\text{I}), \tag{36}$$

$$P_{x_2}(\text{III}) = P_{x_2}(\text{II}). \tag{37}$$

In *Case IV*, both vehicles have to manage the interference. This situation demands that the vehicle with lower power decodes the other vehicle's file, removes it from the NOMA signal, and then decodes its own file. The requirements for successful decoding are as follows:

- For the range $0.5 \leq \alpha \leq 1$:

$$\frac{\alpha PL_1^{\mathcal{B}} |\tilde{h}_1^{\mathcal{B}}|^2}{(1-\alpha) PL_1^{\mathcal{B}} |\tilde{h}_1^{\mathcal{B}}|^2 + \sigma_1^2} \geq \epsilon_1, \tag{38}$$

$$\frac{\alpha PL_2^{\mathcal{B}} |\tilde{h}_2^{\mathcal{B}}|^2}{(1-\alpha) PL_2^{\mathcal{B}} |\tilde{h}_2^{\mathcal{B}}|^2 + \sigma_2^2} \geq \epsilon_1, \tag{39}$$

$$\frac{(1-\alpha) PL_2^{\mathcal{B}} |\tilde{h}_2^{\mathcal{B}}|^2}{\sigma_2^2} \geq \epsilon_2. \tag{40}$$

The success probabilities can be represented as

$$P_{x_1}(\text{IV}_1) = P_{x_1}(\text{II}_1), \tag{41}$$

$$P_{x_2}(\text{IV}_1) = P_{x_2}(\text{I}_1). \tag{42}$$

- For $0 < \alpha < 0.5$:

$$\frac{(1-\alpha) PL_1^{\mathcal{B}} |\tilde{h}_1^{\mathcal{B}}|^2}{\alpha PL_1^{\mathcal{B}} |\tilde{h}_1^{\mathcal{B}}|^2 + \sigma_1^2} \geq \epsilon_2, \tag{43}$$

$$\frac{\alpha PL_1^{\mathcal{B}} |\tilde{h}_1^{\mathcal{B}}|^2}{\sigma_1^2} \geq \epsilon_1, \tag{44}$$

$$\frac{(1-\alpha) PL_2^{\mathcal{B}} |\tilde{h}_2^{\mathcal{B}}|^2}{\alpha PL_2^{\mathcal{B}} |\tilde{h}_2^{\mathcal{B}}|^2 + \sigma_2^2} \geq \epsilon_2. \tag{45}$$

We can present the associated probabilities of successful decoding in the following manner:

$$P_{x_1}(\text{IV}_2) = P_{x_1}(\text{II}_2), \tag{46}$$

$$P_{x_2}(\text{IV}_2) = P_{x_2}(\text{I}_2). \tag{47}$$

### 4.2. Power Allocation

This subsection outlines the power allocation problems for all scenarios under consideration, emphasizing that the likelihood of successful decoding for $x_1$ and $x_2$ is given by the product of their individual success probabilities.

In *Case I*, the probability of successful decoding is contingent on $\alpha$. Hence, the optimization problem for this case is expressed as

$$\mathcal{P}(\text{I}_1) : \max \ P_{x_1}(\text{I}) P_{x_2}(\text{I}_1)$$
$$\text{s.t. } 0.5 \leq \alpha \leq 1. \tag{48}$$

The convexity of $\mathcal{P}(\text{I}_1)$ is significantly influenced by the convexity of $\Phi_k(\frac{\epsilon_1}{\alpha \rho_1 L_1^{\mathcal{B}}})$. We now illustrate the concavity of the objective function in $\mathcal{P}(\text{I}_1)$ within the $0.5 \leq \alpha \leq 1$

interval by providing the following proposition. The concavity of the power allocation problems in other cases can be analogously demonstrated.

**Proposition 1.** *The objection function in $\mathcal{P}(\mathrm{I}_1)$ is concave with respect to $0.5 \leq \alpha \leq 1$.*

**Proof.** Recall that

$$\frac{d^u}{dz^u} G_{p,q}^{m,n}\left(\frac{1}{z}\left|\begin{matrix}a_1,\ldots,a_n,a_{n+1},\ldots,a_p\\b_1,\ldots,b_m,b_{m+1},\ldots,b_q\end{matrix}\right.\right) = (-1)^u z^u G_{p+1,q+1}^{m,n+1}\left(\frac{1}{z}\left|\begin{matrix}1-u,a_1,\ldots,a_n,a_{n+1},\ldots,a_p\\b_1,\ldots,b_m,1,b_{m+1},\ldots,b_q\end{matrix}\right.\right), \quad u = 1,2,\ldots \tag{49}$$

Based on this, the term corresponding to $\Phi_k$ in (17) can be expressed as

$$\frac{d\Phi_1\left(\frac{\epsilon_1}{\alpha\rho_1 L_1^{\mathcal{B}}}\right)}{d\alpha} = \frac{-\frac{\Omega_{1,1}^{(\mathcal{B})}\Omega_{2,1}^{(\mathcal{B})}}{m_{1,1}^{(\mathcal{B})}m_{2,1}^{(\mathcal{B})}}\frac{\rho_1 L_1^{\mathcal{B}}}{\epsilon_1}}{\Gamma\left(m_{1,1}^{(\mathcal{B})}\right)\Gamma\left(m_{2,1}^{(\mathcal{B})}\right)} \cdot \alpha G_{2,4}^{2,2}\left(\frac{\Omega_{1,1}^{(\mathcal{B})}\Omega_{2,1}^{(\mathcal{B})}}{m_{1,1}^{(\mathcal{B})}m_{2,1}^{(\mathcal{B})}}\frac{\epsilon_1}{\alpha\rho_1 L_1^{\mathcal{B}}}\left|\begin{matrix}0, & 1\\m_{1,1}^{(\mathcal{B})},m_{2,1}^{(\mathcal{B})},1,0\end{matrix}\right.\right). \tag{50}$$

Now, let us define

$$\lambda_1 = \frac{\Omega_{1,1}^{(\mathcal{B})}\Omega_{2,1}^{(\mathcal{B})}}{m_{1,1}^{(\mathcal{B})}m_{2,1}^{(\mathcal{B})}}\frac{\epsilon_1}{\rho_1 L_1^{\mathcal{B}}}, \tag{51}$$

which leads to

$$G_{2,4}^{2,2}\left(\frac{\lambda_1}{\alpha}\left|\begin{matrix}0,1\\m_{1,1}^{(\mathcal{B})},m_{2,1}^{(\mathcal{B})},1,0\end{matrix}\right.\right) = 2\left(\frac{\lambda_1}{\alpha}\right)^{\frac{m_{1,1}^{(\mathcal{B})}+m_{2,1}^{(\mathcal{B})}}{2}} \cdot \mathcal{K}_{m_{2,1}^{(\mathcal{B})}-m_{1,1}^{(\mathcal{B})}}\left(2\sqrt{\frac{\lambda_1}{\alpha}}\right). \tag{52}$$

Subsequently, we can obtain the second derivative of (50) as

$$\frac{d}{d\alpha}\left[\alpha G_{2,4}^{2,2}\left(\frac{\lambda_1}{\alpha}\left|\begin{matrix}0, & 1\\m_{1,1}^{(\mathcal{B})},m_{2,1}^{(\mathcal{B})},1,0\end{matrix}\right.\right)\right]$$

$$= \left(\frac{\lambda_1}{\alpha}\right)^{\frac{m_{1,1}^{(\mathcal{B})}+m_{2,1}^{(\mathcal{B})}-1}{2}}\left[2\sqrt{\frac{\lambda_1}{\alpha}}\mathcal{K}_{m_{2,1}^{(\mathcal{B})}-m_{1,1}^{(\mathcal{B})}}\left(2\sqrt{\frac{\lambda_1}{\alpha}}\right) + \mathcal{K}_{m_{2,1}^{(\mathcal{B})}-m_{1,1}^{(\mathcal{B})}-1}\left(2\sqrt{\left(\frac{\lambda_1}{\alpha}\right)}\right)\right.$$

$$\left. - \mathcal{K}_{m_{2,1}^{(\mathcal{B})}-m_{1,1}^{(\mathcal{B})}+1}\left(2\sqrt{\left(\frac{\lambda_1}{\alpha}\right)}\right) - \left(m_{2,1}^{(\mathcal{B})}+m_{1,1}^{(\mathcal{B})}\right)\sqrt{\left(\frac{\alpha}{\lambda_1}\right)}\mathcal{K}_{m_{2,1}^{(\mathcal{B})}-m_{1,1}^{(\mathcal{B})}}\left(2\sqrt{\frac{\lambda_1}{\alpha}}\right)\right]$$

$$> 0, \tag{53}$$

where the inequality is derived using the Bessel function identities

$$\mathcal{K}_{-\nu}(x) = \mathcal{K}_\nu(x), \quad \nu = 0,1,\ldots \tag{54}$$

and

$$\mathcal{K}_\mu(x) \geq \mathcal{K}_\nu(x), \quad \nu \geq \mu = 0,1,\ldots, \quad x \in \mathbb{R}^+. \tag{55}$$

Hence, the first and second derivatives of $P_{x_1}(\mathrm{I})$ are negative and positive, respectively, while those of $P_{x_1}(\mathrm{I}_1)$ exhibit different trends for $\alpha \to 0$ and $\alpha \to 1$. Thus, the objective function of $\mathcal{P}(\mathrm{I}_1)$ is negative, which completes the proof. □

Since $\mathcal{P}(\mathrm{I}_1)$ is a concave optimization problem, it can be efficiently solved by employing a bisection search within the feasible $\alpha$ interval.

For $0 < \alpha < 0.5$, the optimization problem is stated as

$$\mathcal{P}(\mathrm{I}_2) : \max \ P_{x_1}(\mathrm{I})P_{x_2}(\mathrm{I}_2)$$
$$\text{s.t. } 0 < \alpha < 0.5. \tag{56}$$

In *Case II*, the individual optimization problems for $\alpha \geq 0.5$ and $\alpha < 0.5$ are written as follows:

$$\mathcal{P}(\mathrm{II}_1) : \max \ P_{x_1}(\mathrm{II}_1)P_{x_2}(\mathrm{II})$$
$$\text{s.t. } 0.5 \leq \alpha \leq 1. \tag{57}$$

$$\mathcal{P}(\mathrm{II}_2) : \max \ P_{x_1}(\mathrm{II}_2)P_{x_2}(\mathrm{II})$$
$$\text{s.t. } 0 < \alpha < 0.5. \tag{58}$$

The optimization problem for *Case III* is defined as

$$\mathcal{P}(\mathrm{III}) : \max \ P_{x_1}(\mathrm{III})P_{x_2}(\mathrm{III})$$
$$\text{s.t. } 0 < \alpha < 1. \tag{59}$$

Lastly, the optimization problems for *Case IV*, corresponding to $\alpha \geq 0.5$ and $\alpha < 0.5$, are given by

$$\mathcal{P}(\mathrm{IV}_1) : \max \ P_{x_1}(\mathrm{IV}_1)P_{x_2}(\mathrm{IV}_1)$$
$$\text{s.t. } 0.5 \leq \alpha \leq 1. \tag{60}$$

$$\mathcal{P}(\mathrm{IV}_2) : \max \ P_{x_1}(\mathrm{IV}_2)P_{x_2}(\mathrm{IV}_2)$$
$$\text{s.t. } 0 < \alpha < 0.5. \tag{61}$$

## 5. Power Allocation for Multiantenna Case

In this section, we examine a scenario where the vehicle receives NOMA signals from two antenna arrays, subjected to both LoS and NLoS propagations. This leads to power gains of $x_1$ and $x_2$ with an inherent uncertainty. Here, we presume vehicles $V_1$ and $V_2$ directly decode their intended signals, thereby optimizing the power allocations for $x_1$ and $x_2$ under different caching conditions.

*Case I:* In this scenario, $V_1$ possesses a cache of $f_2$, enabling the decoding of $x_1$ independent of $\alpha$. This event occurs with a probability given by

$$
\begin{aligned}
P_{V_1}(\mathrm{I}) &= \Pr\{\mathrm{SNR}_1^{(x_1)} > \epsilon_1\}) \\
&= \Pr\left\{ \left| \tilde{h}_1^{\mathcal{B}} \right|^2 > \frac{\epsilon_1}{\alpha \rho_1 G_{1,f} L_1^{\mathcal{B}}} \right\} \\
&= \sum_{\mathcal{B}} P_{d_1}(\mathcal{B}) \left[ 1 - \Phi_1 \left( \frac{\epsilon_1}{\alpha \rho_1 G_{1,f} L_1^{\mathcal{B}}} \right) \right].
\end{aligned} \tag{62}
$$

On the other hand, $V_2$ has not cached $f_1$ and considers the $f_1$ signal as interference, resulting in successful decoding of $x_2$ with the following probability:

$$P_{V_2}(\mathrm{I}) = \Pr\{\mathrm{SINR}_2^{(x_2)} > \epsilon_2\}$$

$$= \Pr\left\{ \left| \tilde{h}_2^{\mathcal{B}} \right|^2 > \frac{\epsilon_2}{\left[ G_{2,s}(1-\alpha) - \alpha\epsilon_2 m_{2,f} \right] \rho_2 L_2^{\mathcal{B}}} \right\}$$

$$= \sum_{\mathcal{B}} P_{d_2}(\mathcal{B}) \left[ 1 - \Phi_2 \left( \frac{\epsilon_2}{\left[ G_{2,s}(1-\alpha) - \alpha\epsilon_2 m_{2,f} \right] \rho_2 L_2^{\mathcal{B}}} \right) \right]. \tag{63}$$

*Case II:* This case mirrors *Case I*, as $V_2$ successfully decodes $x_2$ regardless of $\alpha$. The probability of this event is

$$P_{V_2}(\mathrm{II}) = \Pr\left\{ \mathrm{SNR}_2^{(x_2)} > \epsilon_2 \right\}$$

$$= \Pr\left\{ \left| \tilde{h}_2^{\mathcal{B}} \right|^2 > \frac{\epsilon_2}{(1-\alpha)\rho_2 G_{2,s} L_2^{\mathcal{B}}} \right\}$$

$$= \sum_{\mathcal{B}} P_{d_2}(\mathcal{B}) \left[ 1 - \Phi_2 \left( \frac{\epsilon_2}{(1-\alpha)\rho_2 G_{2,s} L_2^{\mathcal{B}}} \right) \right]. \tag{64}$$

Meanwhile, $V_1$ successfully decodes $x_1$ with the following probability:

$$P_{V_1}(\mathrm{II}) = \Pr\left\{ \mathrm{SINR}_1^{(x_1)} > \epsilon_1 \right\}$$

$$= \Pr\left\{ \left| \tilde{h}_1^{\mathcal{B}} \right|^2 > \frac{\epsilon_1}{\left[ \alpha G_{1,f} - (1-\alpha)\epsilon_1 m_{1,s} \right] \rho_1 L_1^{\mathcal{B}}} \right\}$$

$$= \sum_{\mathcal{B}} P_{d_1}(\mathcal{B}) \left[ 1 - \Phi_1 \left( \frac{\epsilon_1}{\left[ \alpha G_{1,f} - (1-\alpha)\epsilon_1 m_{1,s} \right] \rho_1 L_1^{\mathcal{B}}} \right) \right]. \tag{65}$$

*Case III:* In this case, both $V_1$ and $V_2$ have cached each other's files, allowing them to independently decode $x_1$ and $x_2$ by nullifying the signals from $f_2$ and $f_1$, irrespective of the $\alpha$ value. Thus, the probabilities of successful decoding of $x_1$ and $x_2$ at $V_1$ and $V_2$ are, respectively, given by

$$P_{V_1}(\mathrm{III}) = \Pr\{\mathrm{SNR}_1^{(x_1)} > \epsilon_1\}$$

$$= \Pr\left\{ \left| \tilde{h}_1^{\mathcal{B}} \right|^2 > \frac{\epsilon_1}{\alpha\rho_1 G_{1,f} L_1^{\mathcal{B}}} \right\}$$

$$= \sum_{\mathcal{B}} P_{d_1}(\mathcal{B}) \left[ 1 - \Phi_1 \left( \frac{\epsilon_1}{\alpha\rho_1 G_{1,f} L_1^{\mathcal{B}}} \right) \right], \tag{66}$$

and

$$P_{V_2}(\mathrm{III}) = \Pr\{\mathrm{SNR}_2^{(x_2)} > \epsilon_2\}$$

$$= \Pr\left\{ \left| \tilde{h}_2^{\mathcal{B}} \right|^2 > \frac{\epsilon_2}{(1-\alpha)\rho_2 G_{2,s} L_2^{\mathcal{B}}} \right\}$$

$$= \sum_{\mathcal{B}} P_{d_2}(\mathcal{B}) \left[ 1 - \Phi_2 \left( \frac{\epsilon_2}{(1-\alpha)\rho_2 G_{2,s} L_2^{\mathcal{B}}} \right) \right]. \tag{67}$$

*Case IV:* In this scenario, the probability of $V_1$ successfully decoding $x_1$ is

$$P_{V_1}(\text{IV}) = \Pr\{\text{SINR}_1^{(x_1)} > \epsilon_1\}$$

$$= \Pr\left\{\left|\tilde{h}_1^{\mathcal{B}}\right|^2 > \frac{\epsilon_1}{\left[\alpha G_{1,f} - (1-\alpha)\epsilon_1 m_{1,s}\right]\rho_1 L_1^{\mathcal{B}}}\right\}$$

$$= \sum_{\mathcal{B}} P_{d_1}(\mathcal{B})\left[1 - \Phi_1\left(\frac{\epsilon_1}{\left[\alpha G_{1,f} - (1-\alpha)\epsilon_1 m_{1,s}\right]\rho_1 L_1^{\mathcal{B}}}\right)\right], \tag{68}$$

while the probability of $V_2$ successfully decoding $x_2$ is expressed as

$$P_{V_2}(\text{IV}) = \Pr\{\text{SINR}_2^{(x_2)} > \epsilon_2\}$$

$$= \Pr\left\{\left|\tilde{h}_2^{\mathcal{B}}\right|^2 > \frac{\epsilon_2}{\left[G_{2,s}(1-\alpha) - \alpha\epsilon_2 m_{2,f}\right]\rho_2 L_2^{\mathcal{B}}}\right\}$$

$$= \sum_{\mathcal{B}} P_{d_2}(\mathcal{B})\left[1 - \Phi_2\left(\frac{\epsilon_2}{\left[G_{2,s}(1-\alpha) - \alpha\epsilon_2 m_{2,f}\right]\rho_2 L_2^{\mathcal{B}}}\right)\right]. \tag{69}$$

Reflecting on the analyses above, we recall that in each case, the probability of successful decoding for both $f_1$ and $f_2$ results from the product of individual success probabilities. Therefore, the optimal power allocation problems in all considered cases coincide with the framework delineated in Section 4. Additionally, referencing the proof of Proposition 1, we can verify the concavity of power allocation problems. However, the corresponding details, while extensive, are left out here for brevity.

## 6. Performance Evaluation

This section presents a numerical analysis considering the parameters delineated in Table 2, unless explicitly mentioned otherwise. We assume the beamsteering error adheres to a Gaussian distribution, characterized by a mean of zero and variance of $\Delta^2$. This leads to the expression $F_{|\mathcal{E}|}(x) = \text{erf}(x/(\Delta\sqrt{2}))$, where $\text{erf}(\cdot)$ represents the error function [21]. Moreover, we uniformly apply the blockage model, $P_{\text{LoS}}(d_{ij}) = e^{-d_{ij}/200}$, in our numerical analysis. We evaluate the effects of several relevant parameters, namely caching conditions $T$ and $\zeta$, and Nakagami parameters $m$ and $\Omega$. We achieve this by independently altering each parameter, with the rest conforming to the specifications in Table 2. Under the current caching policy, vehicles cache files from index 1 to $T$, stopping only upon reaching maximum capacity.

**Table 2.** Default simulation parameters [19,28].

| Coefficients | Values |
| --- | --- |
| Main lobe gains $(M_{1,f}, M_{2,s})$ | $(15, 15)$ dB |
| Side lobe gains $(m_{1,s}, m_{2,f}, m_{1,f}, m_{2,s})$ | $(-5, -5, -5, -5)$ dB |
| Transmit power of BS, $P$ | 0 dBm |
| Noise power, $\sigma_k^2$ | $-94$ dBm |
| Nakagami fading parameters $(m_1^{\mathcal{B}}, m_2^{\mathcal{B}}, \Omega_1^{\mathcal{B}}, \Omega_2^{\mathcal{B}})$ | $(1, 1, 2, 2)$ |
| Array beamwidths $(\theta_1, \theta_2)$ | $(30°, 30°)$ |
| Beamsteering error parameter, $\Delta$ | $5°$ |
| Link lengths $(d_1, d_2)$ | $(25, 27)$ m |
| Path loss exponents $(\alpha_{\text{L}}, \alpha_{\text{N}})$ | $(2, 4)$ |
| Path loss intercepts $(C_{\text{L}}, C_{\text{N}})$ | $(10^{-7}, 10^{-7})$ |
| Caching files of BS, $T$ | 20 |
| Caching size at vehicle, $k$ | 5 |
| Skewness control parameter, $\zeta$ | 0.5 |
| SINR threshold, $\epsilon_k$ | 1 dB |

### 6.1. Single-Antenna Case

Figure 3 depicts the objective function values for optimization problems $\mathcal{P}(\mathrm{I}) \sim \mathcal{P}(\mathrm{IV})$ as they relate to the power allocation factor $\alpha$. As expected, the objective function demonstrates concavity in relation to $\alpha$ in each instance. More specifically, in *Case I*, and in line with Proposition 1, the probability of successful decoding shows concavity for $\alpha \in (0, 0.5)$ and $\alpha \in (0.5, 1)$. Thus, the optimal power allocation $\alpha$ for each instance and its subinstances can be established through the bisection method.

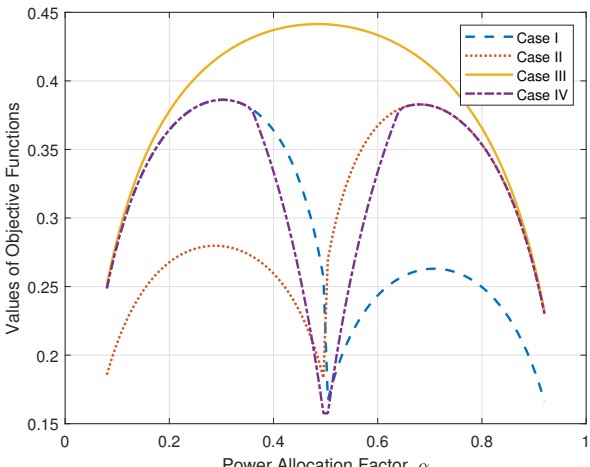

**Figure 3.** The optimization problems exhibit concave characteristics in their objective function values.

The performance comparison between cache-assisted NOMA and OMA schemes, as well as their traditional counterparts, is presented in Figure 4. These cache-assisted approaches exhibit superior success probabilities for decoding compared to their non-cache-assisted counterparts, particularly in scenarios characterized by a low transmit SNR. Notably, the cache-assisted NOMA scheme outperforms its OMA counterpart. Furthermore, Figure 4 illustrates the opportunity to enhance the success probability of decoding for each vehicle by allocating a larger transmit power from the BS.

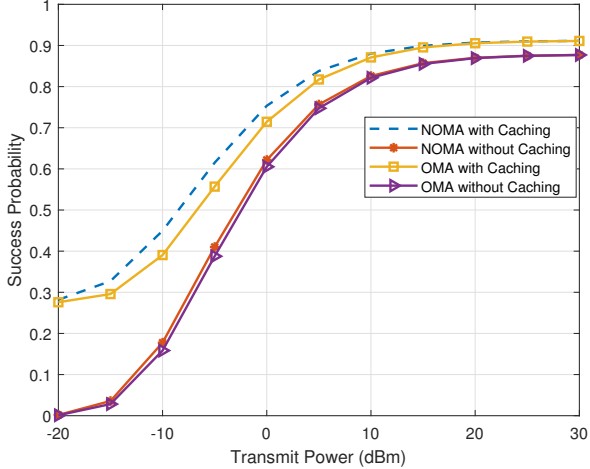

**Figure 4.** Performance comparison of different schemes.

Figure 5 investigates the effect of fading parameters on the proposed cache-assisted NOMA and OMA schemes. Specifically, Figure 5a plots the variations in performance tied to different values of the Nakagami parameter, $m$. As predicted, the success probability associated with cache-assisted NOMA and OMA schemes escalates as the quality of channel conditions ameliorates, achieved through augmenting the Nakagami fading parameter $m$. The proposed scheme consistently shows superior performance to the cache-assisted OMA scheme across a spectrum of fading scenarios. The influence of the Nakagami parameter

$\Omega$ is demonstrated in Figure 5b. It is observed that the success probability increases as $\Omega$ rises. For a range of $\Omega$ values, the proposed scheme sustains a higher success probability than the cache-assisted OMA scheme. Additionally, a more pronounced performance gap between cache-assisted NOMA and OMA schemes is observable in low-SNR regions when larger values of $\Omega$ are employed.

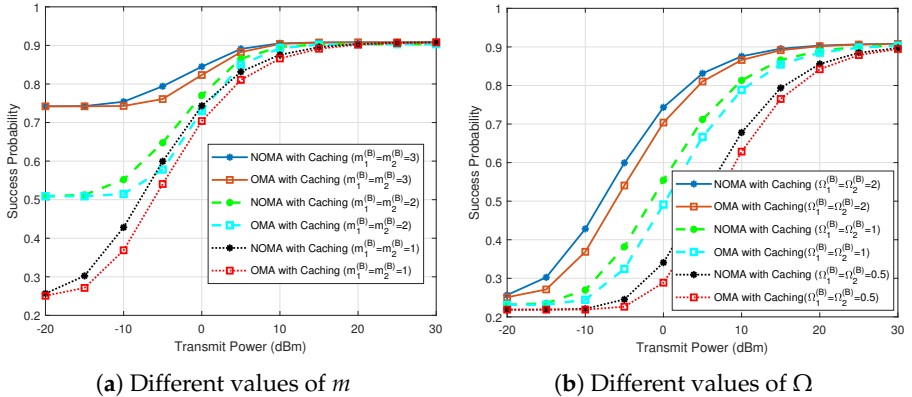

(**a**) Different values of $m$　　　　　　　　　　(**b**) Different values of $\Omega$

**Figure 5.** Effect of Nakagami parameters.

### 6.2. Multiantenna Case

Figure 6 demonstrates the decoding performance of various schemes in a multiantenna scenario, plotting the average probability of successful decoding by NOMA vehicles of their targeted signals as a function of the BS's transmit power. As the BS's transmit power increases, the success probability correspondingly rises. Concurrently, the performance differential among various schemes becomes more evident compared to those in Figure 3. This is attributable to the antenna array offsetting the severe path loss characteristic of mmWave channels, thereby enhancing the SINR of both vehicles' desired signals.

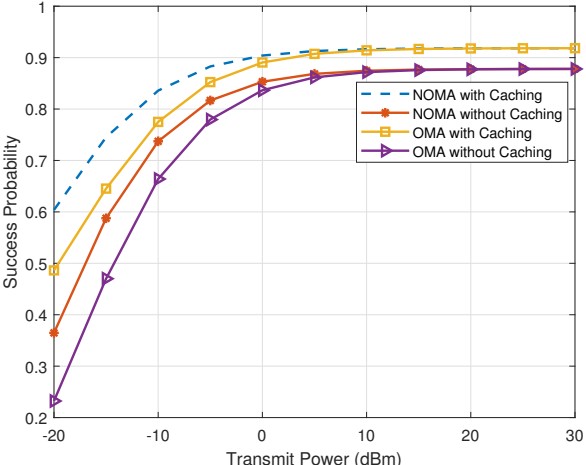

**Figure 6.** Performance comparison of different schemes.

The effect of beamsteering error on the decoding performance of the proposed cache-assisted NOMA scheme and its benchmark counterparts is examined in Figure 7, which depicts the decoding success probabilities as a function of different $\Delta$ values. It should be emphasized that the primary lobe gains, $G_{1,f}$ and $G_{2,s}$, are influenced by beamsteering error, and the BS's transmit power is set to $-10$ dBm. As illustrated in Figure 7, the proposed cache-assisted scheme exhibits increased robustness in terms of beamforming orientation accuracy, attributable to the power allocation optimization across disparate caching scenarios.

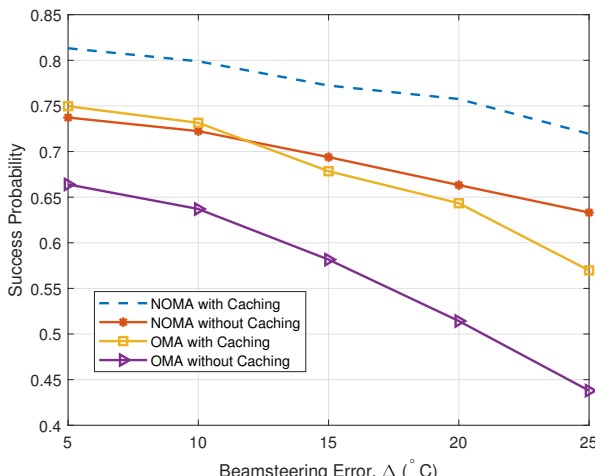

**Figure 7.** Effect of beamsteering error.

Figure 8 depicts the decoding success probabilities under varying caching conditions, taking into account the Zipf distribution parameter $\zeta$ and the caching capacity of the vehicle. To shed light on the impact of caching files on the success probability, the BS's transmit power is held constant at $-10$ dBm. In Figure 8, it is assumed that the distribution of vehicle requests follows the Zipf distribution, distinguished by a certain level of skewness. With an increasing skewness factor $\zeta$, vehicle requests become more concentrated on files with smaller indices. A notable observation from Figure 8 is that NOMA outperforms OMA. In both NOMA and OMA schemes, performance improves with larger vehicle cache size; however, the degree of improvement is more pronounced with a smaller cache size.

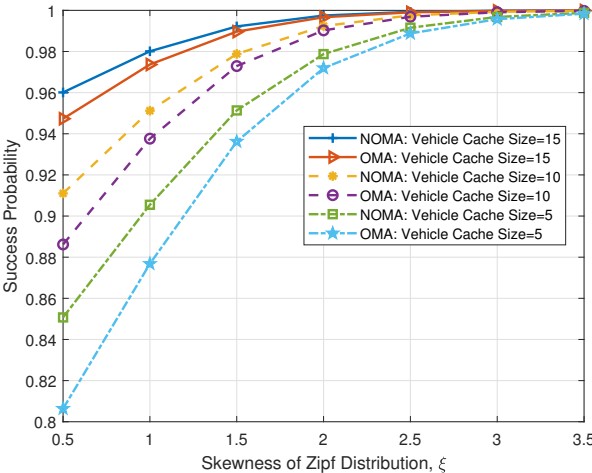

**Figure 8.** Effect of caching conditions.

## 7. Conclusions

In this paper, a cache-assisted NOMA scheme is proposed for mmWave vehicular networks, where a multiantenna BS transmits NOMA signals to a pair of vehicles. Our approach judiciously leverages caching and NOMA techniques to enhance augment both decoding performance and user fairness, simultaneously utilizing the mmWave beam to enhance channel conditions. A thorough analysis was conducted to evaluate the successful decoding probabilities for paired vehicles across various caching scenarios. Under each unique caching condition, an optimal power allocation strategy was devised to maximize the product of individual success decoding probabilities. Through numerical results, the decoding performance of both our proposed scheme and the benchmark schemes were evaluated, thereby explicitly validating the superior performance of our proposed cache-assisted NOMA scheme, especially in the face of beamsteering errors.

**Author Contributions:** Conceptualization, W.C. and J.G.; methodology, W.C. and J.G.; software, W.C.; validation, X.G.; formal analysis, W.C. and J.G.; investigation, G.Z.; writing—original draft preparation, W.C., J.G. and X.G.; writing—review and editing, X.G. and G.Z.; funding acquisition, J.G. and G.Z. All authors have read and agreed to the published version of the manuscript.

**Funding:** This research was funded by the National Natural Science Foundation of China under Grant 61971245, supported by Class C project funded by the 16th batch of 'six talent peaks' high level talent selection and training in Jiangsu Province, grant number XYDXX-245, and sponsored by Qing Lan Project in Jiangsu Province.

**Data Availability Statement:** The authors approve that data used to support the findings of this study are included in the article.

**Conflicts of Interest:** The authors declare no conflict of interest.

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
