# Peer review of "Beamsteering-Aware Power Allocation for Cache-Assisted NOMA mmWave Vehicular Networks"

_electronics, doi:10.3390/electronics12122653_

Round 1
Reviewer 1 Report
Please see the attached file.

Minor editing of English language is needed.
Reviewer 2 Report
The author proposes an optimal power control in cache-assisted NOMA millimeter-wave vehicular networks, including double-Nakagami fading channels, beam-steering errors, various caching scenarios, the goal of maximizing successful signal decoding probability for paired vehicles, enhancing cache utility and NOMA efficiency, while considering fairness, robustness against beam-steering errors, and larger cache sizes. After reviewing the paper, I identified several issues listed in the section below:
1) In the system model's section "MmWave Channel Model", the author has defined two parameters, dk and hk where the author represented the distance between vehicle k and the BS with the symbol dk; however, the author has not described the term hk related to both channels. Moreover, in the determined function expression, the variable C refers to which parameter? Similarly, the omega in equation 4 has not been described in the paper. The author has utilized the modified Bessel function in the distribution function in Eq. 4, however, the reason for using the Bassel is not known.
2) In the cache model associated with the BS, the Zipf distribution with a skewness control parameter, the author needs to describe the reason why the Zipf distribution is adopted along with a detailed discussion of each parameter utilized there.
3) The author has presented the values of the objective function with respect to the power allocation factor; whereas, other dependent parameters have not been evaluated. In the evaluation the comparative study presents only performance comparison with the OMA model only, can the author show some more comparisons to show the superiority of the proposed model with its associated cache?
4) In problem modeling only two vehicles and a BS are considered; however, in real scenarios, the vehicle count is random and could be more than two vehicles communicating the same BS simultaneously. Also, the BS can hope devices/vehicles based on the received signal strengths to any other neighboring BS. The author has not considered complex scenarios to evaluate the performance of the proposed model. Can the author show the NoMa-cache performance for a more complex case?
Moderate editing of English language required.
Reviewer 3 Report
The article deals with optimal power control in cache-assisted NOMA mmWave vehicular networks, where radio channels experience double-Nakagami fading and the mmWave beamforming is subject to beam steering errors.
A comprehensive analysis of the decoding success probabilities under various caching scenarios leads to the development of optimal power allocation strategies for diverse caching conditions.
The proposed technique is proved by numerical results obtained by simulations.
The paper cites 25 sources, some of which are very recent.
The paper can be published with only 2 minor questions/revisions:
1. The name of the first author, after the paper title, should not be capitalized?
2. Fig. 7, x axis label: Beamsteering Error, in °C – Celsius degree?
Round 2
Reviewer 1 Report
The authors have addressed all my concerns, no further comments.
Reviewer 2 Report
The authors solved all my concerns.
Final English check is needed.